# Up-regulation of ubiquitin–proteasome activity upon loss of NatA-dependent N-terminal acetylation

Ilia Kats[1], Christian Reinbold[1], Marc Kschonsak[1] , Anton Khmelinskii[2], Laura Armbruster[1], Thomas Ruppert[1] , Michael Knop[1,3] 

**N-terminal acetylation is a prominent protein modification, and inactivation of N-terminal acetyltransferases (NATs) cause protein homeostasis stress. Using multiplexed protein stability profiling with linear ubiquitin fusions as reporters for the activity of the ubiquitin proteasome system, we observed increased ubiquitin proteasome system activity in NatA, but not NatB or NatC mutants. We find several mechanisms contributing to this behavior. First, NatA-mediated acetylation of the N-terminal ubiquitin–independent degron regulates the abundance of Rpn4, the master regulator of the expression of proteasomal genes. Second, the abundance of several E3 ligases involved in degradation of UFD substrates is increased in cells lacking NatA. Finally, we identify the E3 ligase Tom1 as a novel chain-elongating enzyme (E4) involved in the degradation of linear ubiquitin fusions via the formation of branched K11, K29, and K48 ubiquitin chains, independently of the known E4 ligases involved in UFD, leading to enhanced ubiquitination of the UFD substrates.**

## Introduction

Selective protein degradation is essential for proteome homeostasis, to remove unnecessary or abnormal proteins as part of quality control pathways or in response to changes in the environment. In eukaryotes, the bulk of selective protein degradation is handled by the ubiquitin–proteasome system (UPS). Substrates of the UPS are recognized through features known as degradation signals or degrons (Ravid & Hochstrasser, 2008), ubiquitinated by E3 ubiquitin ligases typically on lysine side chains, and finally degraded by the proteasome (Hershko & Ciechanover, 1998; Finley et al, 2012).

Global activity of the UPS is tightly regulated and responds to environmental challenges such as heat stress, DNA damage, or cytotoxic compounds, which can damage or induce misfolding of proteins (Hahn et al, 2006). In the budding yeast *Saccharomyces cerevisiae*, the transcription factor Rpn4 is a master regulator of proteasome capacity. It trans-activates promoters of all proteasomal subunits and several other proteins of the UPS (Mannhaupt et al, 1999; Shirozu et al, 2015). Expression of Rpn4 is in turn regulated by several stress-induced transcription factors such as Hsf1 and Yap1 (Hahn et al, 2006).

In addition to global regulation of the UPS that affects the entire proteome, selective degradation of specific proteins can be induced through post-translational modifications creating or exposing degradation signals. N-degrons that target for degradation via an N-terminal destabilizing residue can be formed by specific endoproteolytic cleavage. For example, cohesin cleavage by separase at the metaphase–anaphase transition induces degradation of the C-terminal fragment by the Arg/N-end rule pathway that recognizes the newly exposed N-terminal residue as a degradation signal (Rao et al, 2001).

N$\alpha$-terminal acetylation of proteins (Nt-acetylation) is a co-translational modification catalyzed by ribosome-associated N$\alpha$-terminal acetyltransferase (NAT) complexes. Three NATs, NatA, NatB, and NatC, are responsible for the acetylation of 50–90% of all protein N-termini in yeast and human cells (Starheim et al, 2012; Aksnes et al, 2016). These NATs differ in their substrate specificity. NatA acetylates the small residues (S,A,V,C,G) after they have been exposed at the N-terminus through cleavage of the initiator methionine (iMet) by methionine aminopeptidases (MetAPs). NatB and NatC acetylate the iMet if it is followed by a polar residue (one of [D,E,N,Q]) or a large hydrophobic residue (one of [F,L,I,W]), respectively. The identity of the first two N-terminal residues is, however, not sufficient to trigger Nt-acetylation, and numerous proteins lack this modification despite being potential NAT substrates according to their primary sequence (Aksnes et al, 2016).

Nt-acetylation has been implicated in a multitude of cellular processes. Deletion of the major N-acetyl transferase genes leads to pleiotropic effects with distinct influences on the physiology and

---

[1]Zentrum für Molekulare Biologie der Universität Heidelberg (ZMBH), DKFZ-ZMBH Alliance, Heidelberg, Germany   [2]Institute of Molecular Biology (IMB), Mainz, Germany
[3]Deutsches Krebsforschungszentrum (DKFZ), DKFZ-ZMBH Alliance, Heidelberg, Germany

Correspondence: m.knop@zmbh.uni-heidelberg.de
Ilia Kats's present address is Computational Genomics and System Genetics, German Cancer Research Center (DKFZ), Heidelberg, Germany
Marc Kschonsak's present address is Department of Structural Biology, Genentech Inc., South San Francisco, CA, USA
Laura Armbruster's present address is Center for Organismal Studies (COS), Heidelberg, Germany

cellular proteostasis of *S. cerevisiae*. For NatA, several individual targets are known where Nt-acetylation functions in mediating protein–protein interactions, prevention of incorrect protein secretion, protein folding, and degradation. This includes key transcriptional regulators as well as protein folding machinery or structural components of the cytoskeleton (Aksnes et al, 2016; Friedrich et al, 2021); therefore, it is not surprising that the attribution of specific functions to Nα-acetylated N-termini is not possible. Another important point is the question to what extent Nt-acetylation is subject to specific regulation, for example, via regulation of the activity of the individual NATs. Whereas there is some evidence from plants that NatA activity can be regulated as a function of drought stress (Linster et al, 2015), in yeast no clear reports about specific regulation of NATs exist. This is consistent with the observation that Nt-acetylation appears to be irreversible and that it is hardly affected by reduced Acetyl-CoA levels (Varland et al, 2018).

Nt-acetylation was proposed to act as a degradation signal (Hwang et al, 2010b; Shemorry et al, 2013) and Nt-acetylated N-termini are thought to be recognized and ubiquitinated by specific E3 ligases of the Ac/N-end rule pathway. The universality of this pathway is debatable because acetylation is not a self-sufficient degron and the involved E3 ligases recognize a broad palette of N-degrons independent on Nt-acetylation (Zattas et al, 2013; Gawron et al, 2016; Kats et al, 2018; Friedrich et al, 2021). Still, Nt-acetylation can be part of N-degrons that contain adjacent sequence motifs (Hwang et al, 2010b; Shemorry et al, 2013). In recognition of the fact that Nt-acetylation is not a general degron, it was finally proposed to refer to "N-terminal degrons," and to avoid the wording "N-end rule" (Varshavsky, 2019), in favor of specific terminology that refers to the individualistic nature of each N-terminal degron. This is even more important, given that accumulating evidence suggests that N-acetylation can fulfill the exact opposite function: as a protein stabilizing modification. First, Nt-acetylation can prevent direct ubiquitylation of the Nα amino group of proteins (Hershko et al, 1984; Kuo et al, 2004; Caron et al, 2005). This may be the underlying mechanism how Nt-acetylation protects the Derlin protein Der1 from degradation by the associated E3 ligase Hrd1 (Zattas et al, 2013). Acetylation can also protect N-termini from non-canonical processing by aminopeptidases, that is, methionine aminopeptidases 1 and 2 (Map1 and Map2), which in the absence of Nt-acetylation, can remove the initiator methionine (iMet) from the nascent chain. This leads to the exposure of the second residue, which in the case of NatB and NatC N-termini will lead to the exposure of an Arg/N-degron that can targets the protein for Ubr1-dependent degradation (Kats et al, 2018; Nguyen et al, 2018).

The yeast genome encodes several linear ubiquitin fusion proteins which serve as a source of free ubiquitin because the N-terminal ubiquitin moiety is usually co-translationally cleaved off by endogenous deubiquitinating enzymes (DUBs) (Amerik & Hochstrasser, 2004). Linear ubiquitin fusions that escape DUB cleavage or that are generated post-translationally by ubiquitination of the Nα group of the first amino acid residue of a protein can be further ubiquitinated by E3 ligases of the ubiquitin-fusion degradation (UFD) pathway using conventional lysine-ε-amino-specific linkage on at least one of the seven lysine residues of the N-terminal ubiquitin moiety and degraded by the proteasome. In yeast, Ufd4 is the major E3 ligase of the UFD pathway (Johnson et al, 1995), whereas

the accessory E3 ligases Ufd2 and Ubr1 promote degradation by acting as chain elongating enzymes (E4 ligases) (Koegl et al, 1999; Hwang et al, 2010a). The UFD pathway is conserved in humans, where it is composed of the Ufd4 ortholog TRIP12 and the Ufd2 orthologs UFD2a and UFD2b (Park et al, 2009). The pathway was first identified in yeast using artificial substrates consisting of linear ubiquitin fusions (Ubi$^{G76V}$) that are resistant to cleavage by DUBs (Johnson et al, 1995). Such UFD substrates were subsequently used as a high-throughput-compatible readout of proteasome activity (Dantuma et al, 2000; Stack et al, 2000). However, endogenous substrates of the UFD pathway have proven difficult to identify, and only few are known to date. Nevertheless, mammalian cells possess the E2 conjugating enzyme Ube2w that monoubiquitinates N-terminal residues if they are followed by an intrinsically disordered sequence (Scaglione et al, 2013; Tatham et al, 2013; Vittal et al, 2014) as well as the E3 ligase LUBAC that assembles linear M1-linked ubiquitin chains and was implicated in immune signaling (Tokunaga et al, 2009; Gerlach et al, 2011; Fiil et al, 2013). However, to the best of our knowledge, the origin of the N-terminal ubiquitin moiety in known endogenous UFD substrates has not been investigated, and all known instances of N-terminal ubiquitination by LUBAC or Ube2w do not induce degradation of the substrate, but rather mediate protein–protein interactions or activate signaling cascades (Rittinger & Ikeda, 2017). N-terminal ubiquitination has been suggested to be regulated by Nt-acetylation, as both modifications involve the same amino group (Caron et al, 2005; McDowell & Philpott, 2013).

We have developed multiplexed protein stability profiling, a quantitative and high-throughput compatible method that enables the degradation profiling of large peptide libraries using FACS and analysis of enriched fractions by deep sequencing (Kats et al, 2018). We used multiplexed protein stability profiling to explore the degron propensity of native and non-native N-termini and a large fraction of the yeast N-termini (N-terminome) (Kats et al, 2018). In this work we explore the influence of NatA on protein degradation in the budding yeast *S. cerevisiae* starting with the observation that artificial UFD substrates are degraded faster in NatA-deficient cells. Using screening and targeted experiments we describe a role for Nt-acetylation on regulation of UPS activity via Rpn4 and we investigate how the abundance of several E3 and E4 ubiquitin ligases is influenced by NatA and how this contributes to UFD. We furthermore identify Tom1 as a novel ubiquitin chain-elongating enzyme (E4) of the UFD pathway and using in vivo and in vitro assays we investigate ubiquitination by Tom1. Altogether, our data provide new insights into the molecular processes governing UPS activity regulation in the absence of NatA activity, emphasizing the importance of NatA for cellular protein homeostasis.

# Results

## NatA affects turnover of UFD substrates

We performed a systematic survey of degrons in protein N-termini using linear ubiquitin fusion reporter constructs (Kats et al, 2018). These reporters consisted of an N-terminal ubiquitin followed by two variable residues (X and Z), a linker sequence (eK) and a tandem

fluorescent protein timer (tFT). The tFT reports on protein stability independently of expression through the intensity ratio of the slow maturing mCherry and the fast maturing sfGFP fluorescent proteins, which increases as a function of protein half-life in steady state (Khmelinskii et al, 2012, 2016). In the course of that study, we observed that reporters with a proline residue immediately following the ubiquitin moiety (Ubi-PZ-tFT reporters) exhibited increased turnover in strains lacking the N-terminal acetyltransferase NatA (Fig 1A), whereas no destabilization was observed in NatB and NatC mutants (see Fig S3 in Kats et al [2018]). The N-terminal ubiquitin moiety is usually co-translationally cleaved by endogenous deubiquiti-nating enzymes (DUBs) (Bachmair et al, 1986), which enables the exposure of non-native amino acid residues at the N-terminus of the reporter protein. However, a proline residue located directly after ubiquitin impairs cleavage of the ubiquitin moiety by DUBs. Such linear ubiquitin fusions are rapidly degraded by the UFD pathway (Johnson et al, 1995), primarily through the action of the ubiquitin E3 ligases Ufd4 and Ubr1 (Hwang et al, 2010a). In contrast, cleaved Ubi-PZ-tFT reporters with an exposed N-terminal proline are stable (Bachmair et al, 1986; Bachmair & Varshavsky, 1989).

To understand how NatA affects turnover of Ubi-PZ-tFT reporters, we first confirmed that these reporters are affected by deletion of the catalytic NatA subunit (*NAA10*) using cycloheximide chase ex-periments. These immunoblots indicated that abundance and/or degradation of an uncleaved Ubi-PP-tFT reporter are influenced by the absence of the catalytic subunit of NatA (Fig 1B). These results can be explained either by accelerated degradation of uncleaved Ubi-PZ-tFT reporters or by impaired DUB activity in the *naa10Δ* mutant. In DUB-impaired cells, a larger fraction of Ubi-PZ-tFT re-porters would remain uncleaved, and rapid degradation of uncleaved Ubi-PZ-tFT reporters by the UFD pathway would account for their apparent destabilization. To distinguish between these possibilities, we investigated turnover of a non-cleavable Ubi$^{G76V}$-tFT reporter, in which the last glycine of ubiquitin is exchanged for valine to completely prevent cleavage by DUBs (Johnson et al, 1992). Degradation of this reporter was inferred from mCherry/sfGFP ratio

as measured by flow cytometry. Stability of the Ubi$^{G76V}$-tFT reporter in wild type yeast was at the lower end of the tFT dynamic range, therefore no clear effect of *NAA10* deletion could be detected by flow cytometry (Fig 1C, pos. 1 & 2, 5 & 6). As expected, this reporter was strongly stabilized in *ufd4Δ* and *ubr1Δ ufd4Δ* cells. Surprisingly however, it was still degraded in these mutants and moreover, it was destabilized upon deletion of *NAA10* (Fig 1C, pos. 3 & 4, 7 & 8). To confirm this, we performed a CHX chase experiment to compare the stability of the Ubi$^{G76V}$-tFT reporter as a function of NatA in the *ubr1Δ ufd4Δ* mutant background. We observed significantly lower protein levels in the *naa10Δ* mutant (Fig S1A), which may not be explained by the observed very weak stability differences. Friedrich et al (2021) reported altered expression/transcription levels in the absence of NatA, which could explain the differences in protein abundance. To investigate this we used qPCR to quantify tFT-mRNA and detected a ~40% reduction of the mRNA levels of the reporter in the absence of NatA (Fig S1B). To better quantify the turnover of the reporter by CHX chase experiments, we repeated the experiment multiple times for better quantification and estimated the half-lives of each replicate in the respective genetic back-ground (Fig S1C). This detected a slight destabilization of the re-porter in the *naa10Δ* mutant that, together with the data from our tFT-assay, indicates a higher turnover rate in the absence of Naa10. To evaluate weather higher abundance of Naa10 might cause a stabilization of the reporter on the other hand, we over-expressed Naa10 under the control of the GAL1-promotor. Comparison of the wild type and the *ubr1Δ ufd4Δ* background in our tFT-assay revealed no stabilization of the reporter in the presence of highly abundant Naa10 but rather a slight desta-bilization in the wild type background (Fig S1D). This suggests that endogenous levels of Naa10 already have the maximum effect on the reporter stability. The results also suggest that NatA-dependent acceleration of UFD substrate turnover is in-dependent of DUB activity and furthermore that accelerated degradation does not involve the canonical E3 ubiquitin ligases implicated in the degradation of such linear ubiquitin fusions.

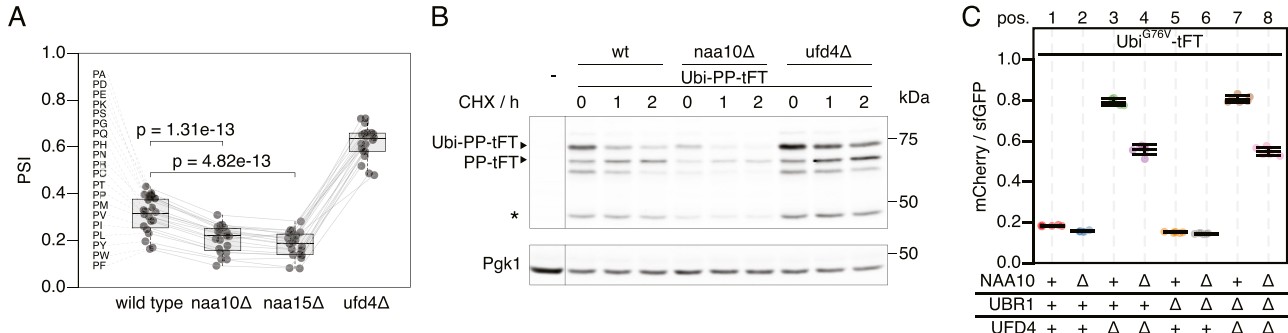

**Figure 1. Accelerated degradation of linear ubiquitin fusion proteins in NatA-deficient strains.**
**(A)** Average stability of Ubi-PZ-tFT reporters in the indicated strains. The protein stability index (PSI) is a measure of protein turnover resulting from high-throughput analysis of tFT-tagged constructs and increases as a function of the mCherry/sfGFP ratio and is therefore anticorrelated with degradation rate. Data from Kats et al (2018). Boxplots show median, first and third quartile, whiskers extend to ± 1.5× interquartile range (IQR) from the box. p: two-sided paired *t* test. **(B)** Degradation of the Ubi-PP-tFT reporter after blocking translation with cycloheximide. Whole-cell extracts were separated by SDS–PAGE followed by immunoblotting with antibodies against GFP and Pgk1 as loading control. A product resulting from mCherry autohydrolysis during cell extract preparation (Gross et al, 2000) is marked (∗). **(C)** Flow cytometry analysis of strains expressing the Ubi$^{G76V}$-tFT reporter. For all flow cytometry experiments, mCherry/sfGFP ratios were normalized to a stable control measured in the same strain background. Mean mCherry/sfGFP ratios and 95% CI of six replicates are plotted together with the median mCherry/sfGFP ratio of each replicate.

## Nt-acetylation by NatA promotes ubiquitin-independent degradation of Rpn4

DUB-independent destabilization of the Ubi[G76V]-tFT reporter in strains lacking the known E3s of the UFD pathway suggested that at least one additional E3 ligase involved in degradation of UFD substrates exists. While searching for this E3, we noticed that deletion of the Ubr2 E3 ligase in the *ubr1Δ ufd4Δ* background accelerated degradation of the Ubi[G76V]-tFT reporter. This destabilization was additive to the effect of *NAA10* deletion on Ubi[G76V]-tFT reporter stability (Fig 2A, pos. 1–4). Ubr2 acts via the Rpn4 transcription factor to regulate expression of UPS genes. More specifically, Rpn4 possesses two degrons, a ubiquitin-dependent degron that is recognized by Ubr2, and an N-terminal ubiquitin-independent degron that is directly recognized by the 26S proteasome (Ju et al, 2004; Ju and Xie, 2004, 2006; Wang et al, 2004a) (Fig 2B). These degrons induce a negative feedback loop regulating UPS activity, such that Rpn4 abundance and consequently proteasome biogenesis are balanced to meet the proteolytic load (Xie & Varshavsky, 2001). Deletion of the Ubr2-dependent degron of Rpn4 (Rpn4Δ(211-229) [Wang et al, 2010]) destabilized the Ubi[G76V]-tFT reporter in the *ubr1Δ ufd4Δ* background. No further destabilization of

this reporter was observed upon additional deletion of *UBR2* (Fig 2A, pos. 5–8). This indicates that accelerated degradation of the Ubi[G76V]-tFT reporter upon ablation of Ubr2 is due to stabilization of Rpn4. Rpn4 is a potential NatA substrate according to its primary sequence, which starts with MA. To explain the additive effect of NatA deletion on degradation of the Ubi[G76V]-tFT reporter, we hypothesized that Nt-acetylation of Rpn4 affects its N-terminal ubiquitin-independent degron. Consistent with this idea, abundance of C-terminally TAP-tagged Rpn4 was strongly increased in the *naa10Δ* mutant (Fig 2C). To test this hypothesis directly, we exploited the portability of the ubiquitin-independent degron of Rpn4 (Ha et al, 2012) and measured turnover of an Rpn4(1-80)-tFT reporter containing the N-terminal Ubi-independent degron of Rpn4 fused to the tFT. This reporter was stabilized upon deletion of *NAA10* (Fig 2D, Pos. 1 & 2). We then substituted the second residue for asparagine to prevent NatA-mediated Nt-acetylation. This strongly reduced stabilization of the reporter in the *naa10Δ* mutant (Fig 2D, compare pos. 3 & 6). Instead, this Rpn4[A2N](1-80)-tFT reporter, which is now a potential target of NatB, was stabilized in a *naa20Δ* NatB mutant (Fig 2D, compare Pos. 4 & 6) to a similar extent as the Rpn4(1-80) reporter in *naa10Δ* cells (Fig 2D, Pos. 1 & 2). These results indicate that Rpn4[A2N](1-80)-tFT is

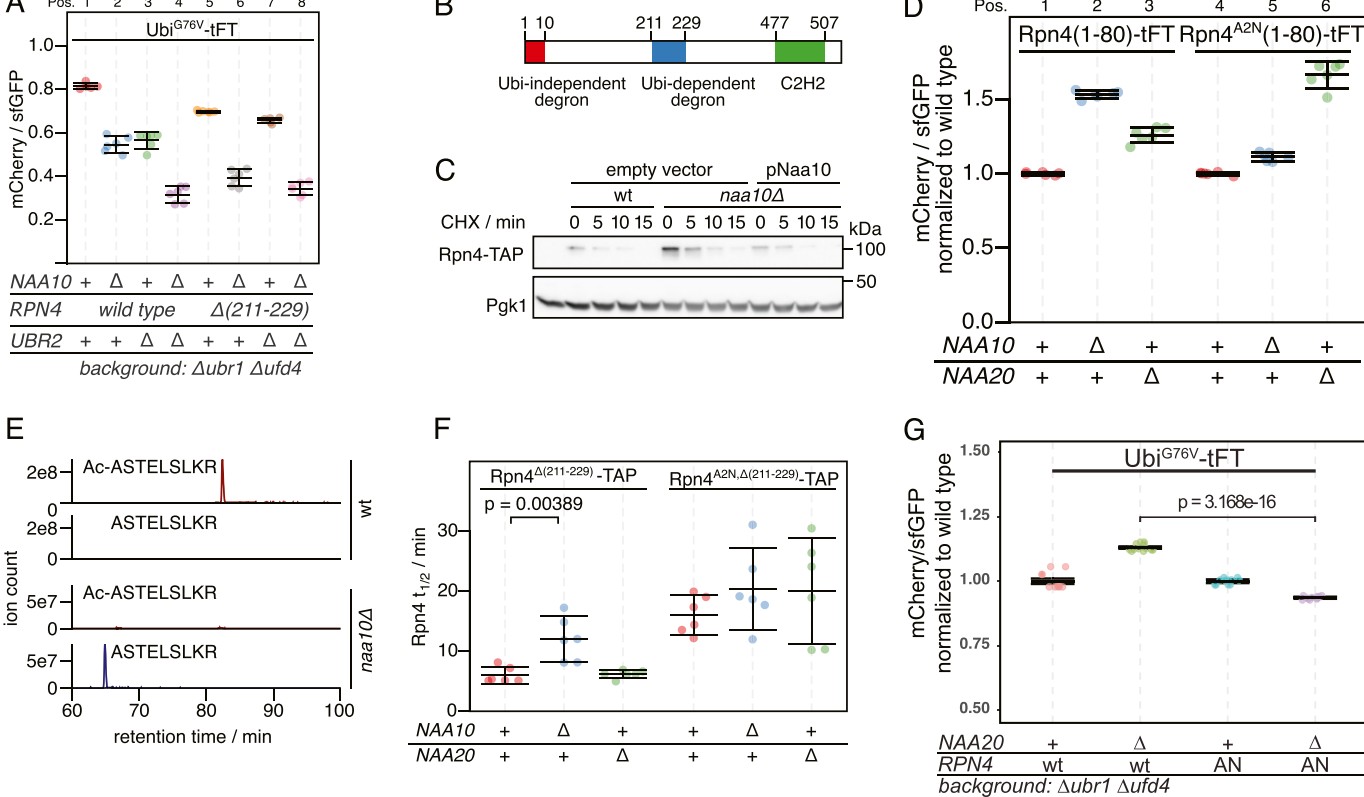

**Figure 2. Regulation of the ubiquitin independent degron of Rpn4 by NatA and contribution to degradation of ubiquitin-fusion degradation substrates.**
**(A)** Flow cytometry analysis of strains expressing the Ubi[G76V]-tFT reporter. **(B)** Domain architecture of Rpn4. **(C)** Degradation of C-terminally TAP-tagged Rpn4 after blocking translation with cycloheximide. Whole-cell extracts were separated by SDS–PAGE followed by immunoblotting with antibodies against protein A and Pgk1 as loading control. **(D)** Flow cytometry analysis of strains expressing the indicated Rpn4 N-terminal sequences fused to the tFT. mCherry/sfGFP ratios were normalized to the mean mCherry/sfGFP ratio of the wild type strain. **(E)** Extracted ion chromatograms of Nt-acetylated and unmodified N-terminal peptides derived from full-length Rpn4 variants obtained by label-free mass spectrometry. **(F)** Half-lives of C-terminally TAP-tagged Rpn4 variants estimated by cycloheximide chase. Mean half-lives and 95% CI of six replicates are plotted together with the half-life of each replicate. p: one-sided unpaired *t* test. **(G)** Flow cytometry analysis of strains expressing the Ubi[G76V]-tFT reporter. mCherry/sfGFP ratios were normalized to the mean mCherry/sfGFP ratio of *ubr1Δ ufd4Δ* cells. AN: Rpn4[A2N]. p: one-sided unpaired *t* test.

acetylated by NatB, and that Nt-acetylation, regardless of the NAT, promotes ubiquitin-independent degradation of Rpn4. We used label-free quantitative mass spectrometry of full-length Rpn4 and could confirm the NatA-dependent acetylation of Rpn4 and NatB-dependent acetylation of Rpn4$^{A2N}$ (Figs 2E and S2C), as expected from the N-terminal sequence of these mutants.

Next, we investigated the influence of NatA on ubiquitin-independent degradation of Rpn4 in a physiological context. We performed cycloheximide chases of C-terminally TAP-tagged Rpn4 lacking its ubiquitin-dependent degron (Rpn4$^{\Delta(211-229)}$–TAP). Deletion of *NAA10* doubled the half-life of Rpn4$^{\Delta(211-229)}$–TAP, but not of Rpn4$^{A2N,\Delta(211-229)}$–TAP (Figs 2F and S2A and B). Similarly, we expected that inactivation of NatB should also lead to a stabilization of Rpn4$^{A2N,\Delta(211-229)}$–TAP, but the results were less clear because of large variation between individual replicates, probably because of the rather severe growth defect caused by the absence of NatB. Nevertheless, our results altogether suggest that NatA-mediated N-terminal acetylation of Rpn4 promotes its ubiquitin-independent degradation, thereby modulating its abundance.

To assess if N-acetylation of Rpn4 mediates the accelerated degradation of the Ubi$^{G76V}$-tFT reporter, we measured turnover of this reporter in cells *naa20Δ* mutant cells expressing wild type Rpn4 or the Rpn4$^{A2N}$ mutant. We found a weak acceleration of degradation of the reporter in the Rpn4$^{A2N}$/*naa20Δ* background compared to the wild type-Rpn4/*naa20Δ* background (Fig 2G). This is consistent with the idea that N-terminal Rpn4 acetylation contributes to the degradation of UFD substrates.

## Tom1 is an E4 ligase of the UFD pathway

Rpn4-independent destabilization of the Ubi$^{G76V}$-tFT reporter in *ubr1Δ ufd4Δ* cells upon deletion of NatB (Fig 2G) is consistent with our initial hypothesis, the existence of an unknown E3 ligase targeting this reporter for degradation. In human cells, the E3 ligase HUWE1 was implicated in the UFD pathway (Poulsen et al, 2012). The yeast homolog Tom1 targets excess histones (Singh et al, 2012), ribosomal subunits (Sung et al, 2016) and other proteins (Kim et al, 2012; Kim & Koepp, 2012) for degradation, but has not yet been described to mediate UFD. We used flow cytometry to test if Tom1 participates in the degradation of UFD substrates and observed only weak stabilization of the Ubi$^{G76V}$-tFT reporter in the *tom1Δ* mutant (Fig 3A, Pos. 1 & 2, for CHX-chase see Fig S5A). This could explain why Tom1 was not previously identified as a component of the UFD pathway. Nevertheless, we were able to co-immunoprecipitate C-terminally TAP-tagged Tom1 with the Ubi$^{G76V}$-tFT reporter (Fig 3B), suggesting a direct role for Tom1 in degradation of UFD substrates. A weak binding was also visible for the Ubi-EH reporter, but no degradation was mediated by Tom1 (Fig S5B) highlighting the specificity of Tom1 to UFD-substrates.

According to the current model of the UFD pathway, linear ubiquitin fusions are first oligoubiquitinated by Ufd4 on the K29 residue of the N-terminal ubiquitin moiety (Johnson et al, 1995; Tsuchiya et al, 2013). These short chains are then extended by the chain-elongating E4 enzymes Ufd2 and Ubr1 to degradation-promoting length (Koegl et al, 1999; Hwang et al, 2010a). Whereas Ubr1 activity has not been investigated in detail, Ufd2 is known to require K48 of the N-terminal ubiquitin moiety (Johnson et al, 1995; Koegl et al, 1999; Liu et al, 2017). The weak stabilization of the Ubi$^{G76V}$-tFT reporter in the *tom1Δ* mutant suggests that Tom1 is redundant with Ufd4 or one of the E4 ligases. UFD substrates lacking K29 are fully stable (Johnson et al, 1995) and thus cannot be used to distinguish between these possibilities. To more confidently place Tom1 in the UFD pathway, we therefore mutated K48 of the Ubi$^{G76V}$-tFT reporter to arginine and measured turnover of the resulting Ubi$^{K48R,G76V}$-tFT reporter using flow cytometry. In wild type yeast, the Ubi$^{K48R,G76V}$-tFT reporter was degraded slower than the Ubi$^{G76V}$-tFT reporter and was not stabilized in a *ufd2Δ* mutant, consistent with the current model. Strikingly, deletion of *TOM1* almost completely abolished degradation of the Ubi$^{K48R,G76V}$-tFT reporter (Fig 3A, Pos. 8 & 9, CHX-chase see Fig S5A) and the *tom1Δ* and *ubr1Δ ufd4Δ* mutants were indistinguishable in terms of Ubi$^{K48R,G76V}$-tFT turnover (Fig 3A, Pos. 9 & 10). Interestingly, the Ubi$^{K48R,G76V}$-tFT reporter was slightly more stable in a *tom1Δ ubr1Δ ufd4Δ* mutant compared to either *tom1Δ* or *ubr1Δ ufd4Δ* cells (Fig 3A, Pos. 9–11). Altogether, these observations argue that Tom1 can play a major role in degradation of UFD substrates. However, Tom1 is not essential for degradation of UFD substrates, as other ubiquitin ligases can use K48 to promote degradation of UFD substrates independently of Tom1. One such ligase is Ufd2, but it is likely that additional ligases performing this function exist, as the Ubi$^{G76V}$-tFT reporter was still degraded in a *tom1Δ ufd2Δ* mutant (Fig 3A).

We considered two mechanisms that could explain our results: (i) UFD substrates are ubiquitinated sequentially by Ufd4 and Tom1 and ubiquitination by Tom1 depends on Ufd4; or (ii) Tom1 ubiquitinates UFD substrates independently of Ufd4 on a lysine residue distinct from K48. In the absence of E4 activity involving K48, ubiquitination by either Ufd4 or Tom1 alone is not sufficient to target the reporter for efficient degradation and both E3 ligases are required. To distinguish between these possibilities and to investigate the effect of Tom1 on ubiquitin

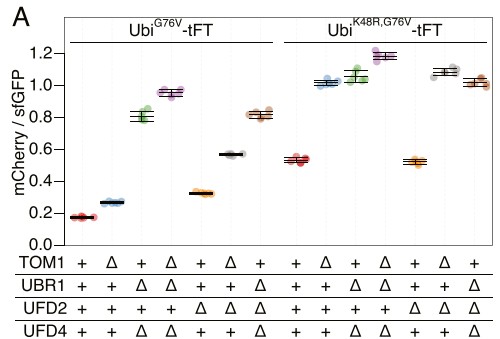

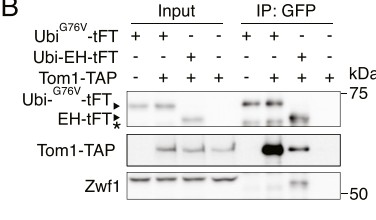

**Figure 3. Role of Tom1 in degradation of ubiquitin-fusion degradation substrates.**
**(A)** Flow cytometry analysis of strains expressing Ubi$^{G76V}$-tFT or Ubi$^{K48R,G76V}$-tFT reporters. **(B)** Co-purification of Tom1 with the Ubi$^{G76V}$-tFT reporter. Proteins were separated by SDS–PAGE followed by immunoblotting with antibodies against GFP, protein A, and Zwf1 as loading control. Input: whole-cell extract prepared by glass bead lysis. IP: proteins eluted after incubation of whole-cell extracts with GFP binding protein coupled to sepharose beads. The EH reporter is not a ubiquitin-fusion degradation substrate. It is therefore thought to not be targeted by Tom1 and served as negative control. (∗) marks a nonspecific band.

chain formation, we purified ubiquitin conjugates from whole-cell extracts. The abundance of high molecular weight species originating from the Ubi$^{G76V}$-tFT reporter was reduced in the *tom1Δ* mutant (Fig 4A, compare lanes 9 & 13). Moreover, mainly mono- and diubiquitinated species were seen in the *tom1Δ* mutant, when using the the Ubi$^{K48R,G76V}$-tFT reporter, despite strong polyubiquitination of this reporter in wild type cells (Fig 4B, compare lanes 5 & 7). In a *ubr1Δ ufd4Δ* background, the Ubi$^{K48R,G76V}$-tFT reporter was only weakly ubiquitinated (Fig 4B, lane 6). Altogether, these results are consistent with the idea that Tom1 acts as a chain elongating enzyme (E4) in the UFD pathway, which recognizes proteins that carry linear oligoubiquitin chains added by Ufd4 and extends these to a degradation-promoting length.

To test this hypothesis directly, we reconstituted ubiquitination of UFD substrates in vitro (Koegl et al, 1999; Hwang et al, 2010a). We first investigated ubiquitination by Ufd4, Ufd2, and Ubr1. Using Ubi-ProtA as a substrate, Ubr1 or Ufd4 alone generated short ubiquitin chains of up to three or four ubiquitin monomers in length, respectively, whereas Ufd2 was inactive in the absence of other E3 ligases (Fig 4C, lanes 1–5). On the other hand, Ufd4 combined with Ufd2 and/or Ubr1 generated high molecular weight conjugates (Fig 4C, lanes 6–8). When Ubi$^{K48R}$-ProtA was used as a substrate, the combination of Ufd4 and Ufd2 did not synthesize appreciable amounts of polyubiquitin conjugates (Fig 4D, compare lanes 3 and 7). Altogether, these results reproduce previous observations

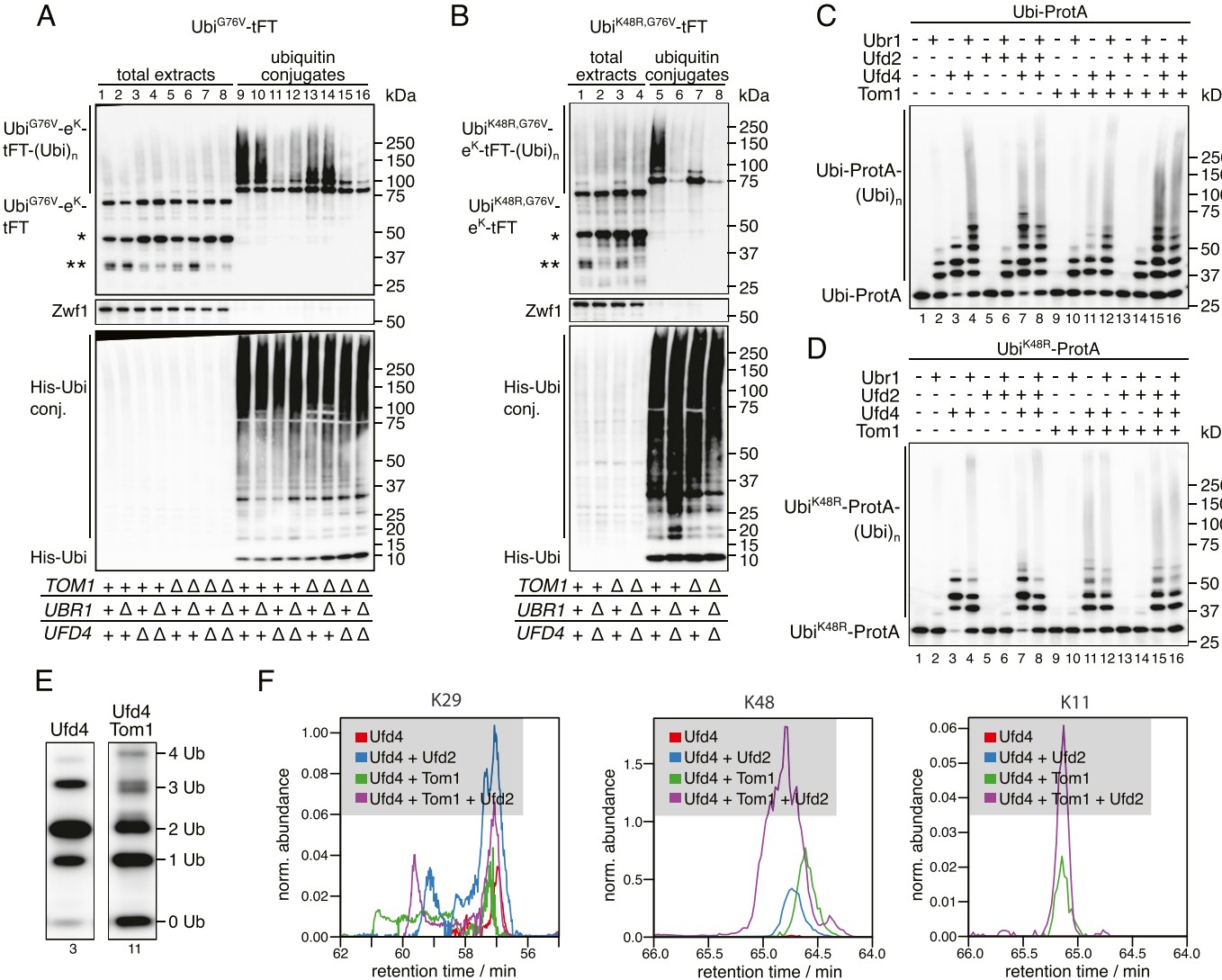

**Figure 4. Tom1 is an E4 ubiquitin ligase and catalyzes the formation of K48 and K11 ubiquitin linkages.**
**(A, B)** Ubiquitination of Ubi$^{G76V}$-tFT (A) or Ubi$^{K48R,G76V}$-tFT (B) in strains expressing 10xHis-tagged ubiquitin. Total cell extracts and ubiquitin conjugates purified by immobilized metal affinity chromatography were analyzed by SDS–PAGE followed by immunoblotting against GFP, Zwf1, and ubiquitin. A product of mCherry hydrolysis during cell extract preparation (Gross et al, 2000) (∗) and a product resulting from inefficient proteasomal degradation of sfGFP (Khmelinskii et al, 2016) (∗∗) are marked. **(C, D)** In vitro reconstitution of ubiquitin chain formation with Ubi-ProtA (C) or Ubi$^{K48R}$-ProtA (D) as substrate using immunoblotting against protein A. **(D, E)** Comparison of the banding pattern of lanes 3 and 11 from (D). Length of ubiquitin chains is indicated. **(F)** Analysis of ubiquitin linkages by mass spec. **(C)** Ubiquitinated proteins were isolated from SDS–PAGE gels prepared from samples in (C) and analyzed for the presence of branched chains as described in methods. The abundance of characteristic fragments in the eluates is shown. Traces were normalized to the non-modified K63 peptide.

(Hwang et al, 2010a; Koegl et al, 1999) and hence confirm the integrity of our in vitro system. Next, we used this assay to investigate the effect of Tom1 on ubiquitin chain formation. Tom1 alone was inactive towards both Ubi-ProtA and Ubi$^{K48R}$-ProtA, but generated high molecular weight polyubiquitin chains in the presence of Ufd4 regardless of the model substrate (Fig 4C and D, lanes 1, 3, 9, and 11 each). This indicates that Tom1 recognizes oligoubiquitinated UFD substrates and either extends pre-formed chains or synthesizes new chains conjugated directly to the substrate. Chain attachment on the N-terminal ubiquitin moiety itself is done on a different residue than K48. Because HUWE1, the mammalian homologue, was shown to synthesize K6- and K11-linked chains (Michel et al, 2017; Yau et al, 2017), it is possible that Tom1 can use those lysine residues of the N-terminal ubiquitin moiety to initiate new chains. Consistent with this, detailed analysis of the banding pattern revealed that ubiquitin conjugates synthesized by Tom1 and Ufd4 are clearly distinct (Fig 4E). Tri-ubiquitinated species of a slightly smaller molecular weight were only generated in the presence of Tom1 and not Ufd4 alone.

We next used mass spectroscopy to identify the type of linkages formed in the in vitro ubiquitination reactions. In reactions that included Ufd4 alone (Fig 4C, lane 3), only K29 linkages were observed (Fig 4F) as expected (Koegl et al, 1999; Liu et al, 2017). Upon addition of Tom1 a strong signal for K48 linkages was observed (Fig 4F) indicating the formation of elongated chains based on K48 linkages. When we tested the high molecular weight products of full reactions (Fig 4C, lane 15) that included Ufd4, Ufd2, and Tom1 we could also detect K11 linkages, whereas these linkages were absent in this fraction when Tom1 was omitted (Fig 4C, lane 8). Together these results support the idea Tom1 functions as an E4 enzyme and that it is able to form different types of ubiquitin linkages.

Next, we tested if Tom1 contributes to the destabilization of UFD substrates in NatA-deficient cells. Deletion of *NAA10 ubr1Δ ufd4Δ tom1Δ*

cells caused a stronger stabilization of the Ubi$^{G76V}$-tFT reporter compared to deletion of *NAA10* in Tom1-proficient cells carrying the rpn4A2N allele (Fig 5A). These results indicate that accelerated turnover of UFD substrates in the *naa10Δ* mutant is mediated partially by Tom1, partially by reduced ubiquitin-independent degradation of Rpn4, and partially by other factors.

Increased abundance of Tom1 and/or other UFD-specific E3 ligases in the *naa10Δ* background could explain accelerated turnover of UFD substrates in this mutant. Supporting this notion, elevated levels of Ubr1 in NatA-deficient cells have been observed previously (Oh et al, 2017). We therefore tested if NatA affects abundance of Ufd4 and Tom1 using immunoblotting. We observed elevated levels of Tom1 and slightly, but significantly increased Ufd4 abundance in *naa10Δ* cells compared to wild type (Figs 5B and S3A–C).

To test if increased abundance of E3s participating in UFD can accelerate degradation of UFD substrates, we measured degradation of Ubi$^{G76V}$-tFT and Ubi-PZ-tFT reporters in strains overexpressing Ufd4, Tom1, or Ubr1 using flow cytometry. No clear changes in turnover of the Ubi$^{G76V}$ reporter were detected, most likely because it is at the lower limit of the tFT dynamic range in wild type cells. The Ubi-PP-tFT reporter was more stable in the wild type background but was only weakly destabilized in a strain overexpressing Ubr1 (Fig 5C), consistent with a negligible contribution of Ubr1 to UFD in vivo (Figs 1C and 3C [Hwang et al, 2010a]). However, overexpression of Ufd4 or Tom1 strongly destabilized the PP reporter (Fig 5C). Only a fraction of this reporter is degraded by the UFD pathway, whereas the other fraction is stable due to removal of the N-terminal ubiquitin moiety by DUBs. Increased turnover of this reporter upon overexpression of Ufd4 and Tom1 therefore indicates that these E3 ligases can compete with DUB activity. Moreover, these results suggest that increased abundance of UFD E3 ligases could explain accelerated turnover of UFD substrates in the *naa10Δ* mutant.

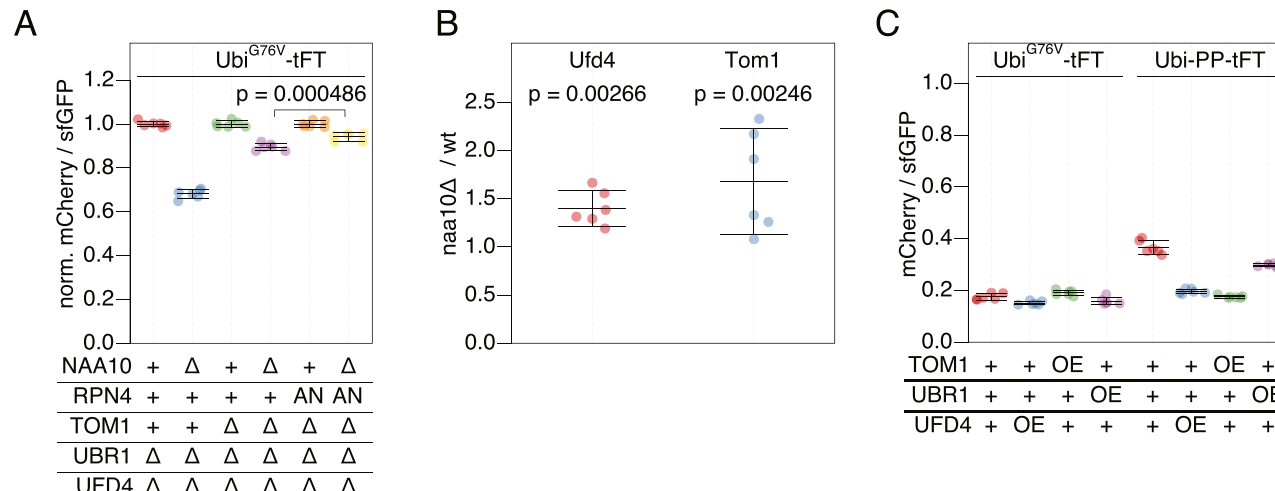

**Figure 5. Role of NatA in regulation of the ubiquitin-fusion degradation.**
**(A)** Flow cytometry analysis of strains expressing the UbiG76V-tFT reporter. mCherry/sfGFP ratios were normalized to the mean mCherry/sfGFP ratio of *NAA10* wild type strains. AN: Rpn4$^{A2N}$. p: one-sided unpaired *t* test. **(B)** Abundance of C-terminally 3HA-tagged Ufd4 or TAP-tagged Tom1 in cells lacking NatA compared to wild type yeast. Whole-cell extracts were separated by SDS–PAGE followed by immunoblotting with antibodies against HA and Pgk1 (Ufd4) or with antibodies against protein A and Fas (Tom1). Pgk1 and Fas served as loading controls. Mean fold-change and 95% CI of six replicates are plotted together with the fold-change of each replicate. p: one-sample *t* test. **(C)** Flow cytometry analysis of strains expressing UbiG76V-tFT or Ubi-PP-tFT reporters. OE, overexpression from the GPD promoter.

# Discussion

Our study sheds light on the impact of NatA Nt-acetylation on protein homeostasis. NatA mutants exhibit specific phenotypes, some of which can be explained by impaired protein–protein interactions in the absence of correctly acetylated N-termini, with various consequences: transcriptional alterations caused by defective Sir3-dependent gene silencing (Wang et al. 2004b), impaired function and stability of the Hsp90 chaperonin system and its client proteins (Oh et al, 2017), cellular sorting of secretory proteins, functions of the Golgi apparatus and the actin cytoskeleton and targeting of specific proteins for degradation (summarized in Aksnes et al [2016]). It is easy to imagine that a multitude of individual effects can challenge proteostasis regulation that then demands for a higher activity of the UPS to remove damage: mistargeted proteins, misfolded proteins, mis-expressed proteins and subunits. This higher UPS activity then could at least partially account for the increased degradation rate of linear ubiquitin fusion proteins.

Interestingly, our observation that Rpn4 Nt-acetylation enhances the strength of its Nt-degron provides a hint towards a more direct coupling of NatA and proteostasis regulation. Here we demonstrate that Nt-acetylation can act independently of E3 ligases to promote ubiquitin-independent degradation of Rpn4, thereby linking NatA activity to regulation of UPS activity. Importantly, in this context Nt-acetylation is neither required nor sufficient to trigger degradation of Rpn4, but rather accelerates degradation of this already unstable protein. Although abundance and half-life of Rpn4 were increased in NatA-deficient cells, we did not observe clearly increased activity of proteasomal subunit promoters (Fig S4). This could be explained by the relatively weak effect of NatA on Rpn4 degradation and abundance, and it is consistent with the previous report that the abundance of proteasomal subunits was not significantly increased even when the N-terminal degron of Rpn4 was completely removed (Wang et al, 2004a), and that it showed only a modest increase in response to expression of a non-degradable Rpn4 variant lacking both degrons (Wang et al, 2010). Because promoters of E3 ligases involved in UFD appear to lack obvious Rpn4-binding motifs (Shirozu et al, 2015), it is unlikely that the increased abundance of Tom1, Ubr4 (Fig 5B) and Ubr1 (Oh et al, 2017) E3 ligases in the *naa10Δ* mutant is mediated by Rpn4. It can be imagined that load-dependent inhibition of autoubiquitination regulates E3 abundance, as shown for other E3 ligases (de Bie and Ciechanover, 2011). Alternatively, Rpn4-independent NatA-mediated regulation of E3 expression is possible.

We furthermore show that degradation of UFD substrates is accelerated in NatA-deficient cells and subsequently identify the E3 ligase Tom1 as a novel E4 chain elongating enzyme of this pathway. This function of Tom1 is clearly distinct from its previously recognized roles as an E3 ligase that is sufficient for ubiquitination of substrate proteins (Sung et al, 2016) and its E3-independent function in ribosome-associated quality control (Defenouillère et al, 2013). Although no endogenous substrates of the UFD pathway are known in yeast, the pathway is conserved in mammalian cells, where several functions have been identified. UBB+1, a mutant ubiquitin variant with a short C-terminal extension caused by a frameshift mutation, is a substrate of the mammalian UFD pathway (Park et al, 2009) and has been linked to neurodegenerative disorders (van

Leeuwen et al, 1998). The cell cycle regulator p21 (Bloom et al, 2003), the ERK3 MAP kinase (Coulombe et al, 2004), and the Arf tumor suppressor (Kuo et al, 2004) were shown to be degraded after N-terminal ubiquitination. It was recently demonstrated that HUWE1, the mammalian ortholog of Tom1, can ubiquitinate MyoD, the first-known UFD substrate, on its N-terminal residue (Noy et al, 2012). Given the conserved nature of UFD and its components, we speculate that Tom1 can generate endogenous UFD substrates in yeast.

Given that deletion of *NAA10* in a *ubr1Δ ufd4Δ tom1Δ rpn4Δ2N* background still slightly accelerated degradation of the Ubi$^{G76V}$-tFT reporter (Fig 5A), we hypothesize that this destabilization is not due to the action of one single protein, but rather the result of a systemic up-regulation of the UPS, caused in part by reduced ubiquitin-independent degradation of Rpn4, but also by other factors currently unknown. A reason for this could be unspecific, low-efficiency ubiquitination of the N-terminal ubiquitin moiety by most, if not all, cellular E3 ligases, in addition to the specific, high-efficiency ubiquitination by Ufd4 and Tom1. Up-regulation of the UPS would therefore lead to not only an increase in specific and unspecific ubiquitination of UFD substrates but also accelerated proteasomal degradation.

Beside the changed degradation rates of UFD-substrates we also observed changed protein abundance in NatA mutants, which were potentially linked to altered mRNA levels (Fig S1A and B). Changed protein synthesis and transcription rates were already reported for NatA mutants and underline the vast cellular changes that are happening in these deletion strains (Friedrich et al, 2021). In some instances, these phenotypes made it difficult to review stability changes in cycloheximide chase experiments when protein levels were very variable between different genetic backgrounds. Nevertheless, combining the results gained from these experiments with our tFT-assay allowed us to detect also weaker degradation rate changes (Figs S1A–D and 1C).

Altogether, our results complement the knowledge about the role of NatA dependent Nt-terminal acetylation and how this is coupled to the activity of the UPS. We believe that our work will contribute to a better understanding of this protein modification and its functions.

# Materials and Methods

### Yeast genome manipulations

Yeast gene deletions and promoter duplications were performed by PCR targeting, as described (Wach et al, 1994; Janke et al, 2004; Huber et al, 2014). Seamless genome editing was performed using the 50:50 technique (Horecka & Davis, 2014). Yeast strains and plasmids used in this study are listed in Tables S1 and S2, respectively.

### tFT-based protein stability measurements with flow cytometry (tFT assay)

Yeast strains containing the desired plasmids were inoculated into 200 µl SC medium lacking the appropriate amino acids for plasmid

selection and grown to saturation in 96-well plates. The cultures were then diluted into fresh medium by pinning to a new 96-well plate using a RoToR pinning robot (Singer Instruments) and incubated at 23°C for 20–24 h to $1 \times 10^6$–$8 \times 10^6$ cells/ml. Flow cytometry was performed on a FACSCanto RUO HTS flow cytometer (BD Biosciences) equipped with a high-throughput sample loader, a 561 nm laser with 600 nm long pass and 610/20 nm band pass filters for mCherry, and a 488 nm laser with 505 nm long pass and 530/30 nm band pass filters for sfGFP. Data analysis was performed in $R$ (R Core Team, 2016) with the flowCore and flowWorkspace packages using a custom script. Briefly, the events were gated for mCherry- and sfGFP-positive cells, the median intensity of a negative control was subtracted from each channel, and the mCherry/sfGFP ratio was calculated for each cell. The median mCherry/sfGFP ratio of each sample was used for further analysis. Unless otherwise stated, each experiment was performed using two biological replicates with three technical replicates each. To account for growth rate differences, sample mCherry/sfGFP ratios were normalized to the stable Ubi-TH-eK-tFT reporter (plasmid pAnB19-TH, Table S2), which was measured in each strain background.

### Flow cytometry of promoter duplications

Yeast cells were inoculated into 200 $\mu$l SC medium and grown to saturation in 96-well plates. The cultures were then diluted into fresh medium by pinning to a new 96-well plate using a RoToR pinning robot (Singer Instruments) and incubated at 23°C for 20–24 h to $1 \times 10^6$–$8 \times 10^6$ cells/ml. Flow cytometry was performed on a FACSCanto RUO HTS flow cytometer (BD Biosciences) equipped with a high-throughput sample loader, a 561 nm laser with 600 nm long pass and 610/20 nm band pass filters for mCherry, and a 488 nm laser with 505 nm long pass and 530/30 nm band pass filters for sfGFP. Data analysis was performed in R (R Core Team, 2016) with the flowCore and flowWorkspace packages using a custom script. Briefly, the events were gated for single cells using forward and side scatter pulse width, followed by gating for fluorescent cells. The median intensity of a negative control was subtracted from each cell. The median sfGFP intensity of each sample was used for further analysis. Unless otherwise stated, each experiment was performed using two biological replicates with three technical replicates each.

### Cycloheximide chases

Cells were grown at 23°C to $6 \times 10^6$–$1 \times 10^7$ cells/ml in synthetic medium before addition of cycloheximide (100 mg/ml stock in 100% ethanol; Sigma-Aldrich) to 100 $\mu$g/ml final concentration. At each time point, 1 ml of the culture was removed, mixed with 150 $\mu$l 1.85 M NaOH and 10 $\mu$l 2-mercaptoethanol and flash-frozen in liquid nitrogen. Protein extracts were prepared as described (Knop et al, 1999), followed by SDS–PAGE and immunoblotting.

For Ubi-P-tFT constructs as well as Figs S1C and S5A, membranes were probed with rabbit anti-GFP (ab6556; Abcam) and mouse anti-Pgk1 (22C5D8; Molecular Probes) antibodies. A secondary donkey anti-mouse antibody coupled to IRDye800 (610-732-002, biomol; Rockland) or donkey anti-rabbit coupled to Alexa 680 (A10043; Life Technologies) were used for detection on an Odyssey infrared imaging system (Li-Cor).

For Rpn4-TAP strains, membranes were probed with rabbit peroxidase-anti-peroxidase (PAP) antibodies (Z0113; Dako) and imaged on an LAS-4000 system (Fuji), followed by probing with mouse anti-Pgk1 (22C5D8; Molecular Probes) and goat anti-mouse HRP (115-035-003; Dianova) antibodies and imaging. Quantification was performed in ImageJ (Schneider et al, 2012).

For HA-tagged Ufd4, membranes were probed with mouse anti-HA (12CA5) and mouse anti-Pgk1 (22C5D8; Molecular Probes), followed by probing with mouse anti-Pgk1 (22C5D8; Molecular Probes) and imaging on a LAS-4000 system (Fuji).

### Tom1 abundance

Cells expressing protein A-tagged Tom1 were grown at 23°C to $6 \times 10^6$–$1 \times 10^7$ cells/ml in synthetic medium and samples were taken and cell extracts were prepared as described (Knop et al, 1999). After SDS–PAGE and Western blotting, membranes were probed with rabbit peroxidase-anti-peroxidase (PAP) antibodies (Z0113; Dako) and imaged on an LAS-4000 system (Fuji), followed by probing with rabbit anti-Fas (Egner et al, 1993) and goat anti-rabbit HRP (111-035-003; Dianova) antibodies and imaging. Quantification was performed in ImageJ (Schneider et al, 2012).

### Rpn4 mass spectrometry

*pdr5Δ ubr2Δ* yeast cells expressing transcriptionally inactive Rpn4C477A mutants (Wang et al, 2004a) C-terminally tagged with 10xHis-sfGFPcp8 (Khmelinskii et al, 2016) from a GPD promoter were grown in SC-His to $7 \times 10^6$–$8 \times 10^6$ cells/ml. Bortezomib was added to 50 $\mu$M and cultures were incubated for 1 h. $2.5 \times 10^9$ cells were harvested by centrifugation, washed with 20% (wt/vol) trichloroacetic acid, and stored at –80°C. Cell pellets were resuspended in 1,600 $\mu$l 20% (wt/vol) trichloroacetic acid and lysed with 0.5 mm glass beads (Sigma-Aldrich) in a FastPrep FP120 (Thermo Fisher Scientific) for $8 \times 40$ s at 6.5 m/s. After precipitation, proteins were washed with cold acetone, air-dried, and resuspended in 3 ml purification buffer (6M guanidium chloride, 100 mM Tris–HCl, pH 9.0, 300 mM NaCl, 10 mM imidazole, and 0.2% [vol/vol] Triton X-100). DTT was added to 10 mM and samples were incubated at 60°C for 30 min, followed by quenching with 100 mM chloroacetamide at RT for 60 min. Lysates were clarified at 21,000$g$, 4°C for 45 min and the supernatants incubated with TALON beads (Clontech) pre-equilibrated with purification buffer at RT overnight with overhead rotation followed by washing with wash buffer (8M urea, 100 mM sodium phosphate, pH 7.0, 300 mM NaCl, 5 mM imidazole, 5 mM chloroacetamide, and 0.2% [vol/vol] Triton X-100) without (twice) and with 0.2% (wt/vol) SDS (twice). Rpn4 was eluted in 30 $\mu$l elution buffer (8M urea, 100 mM sodium phosphate, pH 7.0, 300 mM NaCl, 500 mM imidazole, 5 mM chloroacetamide, and 0.2% [vol/vol] Triton X-100). 7 $\mu$l of eluate were used for SDS–PAGE followed by Coomassie brilliant blue staining. Bands of the expected size were excised, digested with trypsin, and analyzed with ESI LC–MS/MS on a Q-Exactive HF (Thermo Fisher Scientific) coupled with Dionex Ultimate 3000 RSLCnano (Thermo Fisher Scientific). Mass spectrometry was performed at the ZMBH core facility for mass spectrometry and proteomics.

## Ubiquitin pull-downs

Ubiquitinated proteins were purified from yeast cells expressing N-terminally 10xHis-tagged ubiquitin using a protocol adapted from Khmelinskii et al (2014). Yeast were grown in SC-His/Leu to 7 × $10^6$–8 × $10^6$ cells/ml. Approx. 1 × $10^9$ cells were harvested by centrifugation, washed with cold $H_2O$, and stored at –80°C. Cell pellets were resuspended in 800 $\mu$l 20% (wt/vol) trichloroacetic acid and lysed with 0.5 mm glass beads (Sigma-Aldrich) in a FastPrep FP120 (Thermo Fisher Scientific) for 8 × 40 s at 6.5 m/s. After precipitation, proteins were washed with cold acetone, air-dried, resuspended in 1.5 ml purification buffer (6M guanidium chloride, 100 mM Tris–HCl, pH 9.0, 300 mM NaCl, 10 mM imidazole, 5 mM chloroacetamide, and 0.2% [vol/vol] Triton X-100), and clarified at 21,000$g$, 4°C for 45 min. Protein concentration was determined with Bradford assay (Bio-Rad) in purification buffer diluted 1:10 with $H_2O$. 1% of the amount of protein to be used for purification was removed, precipitated with 150 $\mu$l 20% (wt/vol) trichloroacetic acid, and resuspended in 100 $\mu$l HU buffer (8 M Urea, 5% [wt/vol] SDS, 200 mM sodium phosphate, pH 7.0, 1 mM EDTA, and 15 mg/ml DTT) to be used as total extract. Equal amounts of protein were incubated with TALON beads (Clontech) pre-equilibrated with purification buffer at RT for 1 h 30 min with overhead rotation, followed by washing with wash buffer (8M urea, 100 mM sodium phosphate, pH 7.0, 300 mM NaCl, 5 mM imidazole, 5 mM chloroacetamide, and 0.2% [vol/vol] Triton X-100) without (twice) and with 0.2% (wt/vol) SDS (twice). Ubiquitin conjugates were eluted in 50 $\mu$l elution buffer (8M urea, 100 mM sodium phosphate, pH 7.0, 300 mM NaCl, 500 mM imidazole, 5 mM chloroacetamide, and 0.2% [vol/vol] Triton X-100) and analyzed by SDS–PAGE on 4–12% NuPAGE Bis-Tris gradient gels (Invitrogen) followed by immunoblotting. After probing with rabbit anti-GFP (ab6556; Abcam) and rabbit anti-Zwf1 (Miller et al, 2015) followed by goat anti-rabbit IgG-HRP (#111-035-003; Dianova) and imaging on an LAS-4000 system (Fuji), membranes were stripped (100 mM glycine, 2% [wt/vol] SDS, pH 2.0) and re-probed with mouse anti-ubiquitin (P4G7) followed by goat anti-mouse IgG-HRP (#115-035-003; Dianova) and imaging.

## LC–MS analysis of ubiquitin linkages

SDS–PAGE gels of in vitro ubiquitination products (Fig 4C and D) were stained using Coomassie and from each lane the regions corresponding to the polyubiquitinated species were cut out and processed as described with minor modifications (Fecher-Trost et al, 2013). In brief, after reduction with dithiothreitol and alkylation with iodoacetamide, trypsin digestion was carried out overnight at 37°C. The reaction was quenched by addition of 20 $\mu$l of 0.1% trifluoroacetic acid (TFA; Biosolve) and the supernatant was dried in a vacuum concentrator before LC–MS analysis. Nanoflow LC–MS2 analysis was performed with an Ultimate 3000 liquid chromatography system coupled to a QExactive HF mass spectrometer (Thermo Fisher Scientific). Samples were dissolved in 0.1% TFA and injected to a self-packed analytical column (75 $\mu$m × 200 mm; ReproSil Pur 120 C18-AQ; Dr Maisch GmbH) and eluted with a flow rate of 300 nl/min in an acetonitrile-gradient (3–40%). The mass spectrometer was operated in data-dependent acquisition mode, automatically switching between MS and MS2.

Collision induced dissociation MS2 spectra were generated for up to 20 precursors with normalized collision energy of 29%.

### Database search

Raw files were processed using Proteome Discoverer 2.3. (Thermo Fisher Scientific) for peptide identification and quantification. MS2 spectra were searched with the SEQUEST software (Thermo Fisher Scientific) against the Uniprot yeast database (6,910 entries) and the contaminants database (MaxQuant version 1.5.3.30 (Cox & Mann, 2008)) with the following parameters: carbamidomethylation of cysteine residues as fixed modification and acetyl (protein N-term), oxidation (M), deamidation (NQ), and GG signature for ubiquitination (K) as variable modifications, trypsin/P as the proteolytic enzyme with up to two missed cleavages. Peptide identifications were accepted if they could be established at greater than 95.0% probability by the peptide prophet algorithm (Keller et al, 2002). Protein identifications were accepted if they could be established at greater than 95.0% probability and contained at least two identified peptides. Protein probabilities were assigned by the ProteinProphet algorithm (Alexey et al, 2003). Scaffold (version Scaffold_4.8.4, Proteome Software Inc.) was used to validate and visualize MS/MS based peptide and protein identifications. For graphic presentation of XICs retention times were aligned and exported as .csv files using FreeStyle (Thermo Fisher Scientific).

## Tom1 co-immunoprecipitation

Yeast strains expressing the desired construct were grown to 7 × $10^6$–8 × $10^6$ cells/ml. 1 × $10^9$ were harvested by centrifugation, washed with cold $H_2O$, and stored at –80°C. GFP fusions were immunoprecipitated using laboratory-purified GFP binding protein (GBP) (Rothbauer et al, 2008) coupled to NHS-activated Sepharose FastFlow beads (GE Healthcare) using a protocol adapted from Babiano et al (2012). Cell pellets were resuspended in 200 $\mu$l cold lysis buffer (50 mM Tris–HCl pH 7.4, 150 mM $CH_3COOK$, 5 mM EDTA, 5 mM EGTA, and 0.2% Triton X-100) with protease inhibitors (2× Roche Complete EDTAfree, 5 mM benzamidine, 5 mM Pefabloc SC, 5 mM 1,10-phenanthroline, and 25 mM N-ethylmaleimide) and lysed with 0.5 mm glass beads (Sigma-Aldrich) in a FastPrep FP120 for 6 × 20 s at 6.5 m/s. Lysates were clarified at 21,000$g$ for 30 min and the supernatants incubated for 2 h at 4°C with overhead rotation together with 40 $\mu$l GBP-beads previously equilibrated by washing three times with 1 ml lysis buffer. The beads were washed three times with lysis buffer and eluted in 50 $\mu$l HU buffer (8 M Urea, 5% [wt/vol] SDS, 200 mM sodium phosphate, pH 7.0, 1 mM EDTA, and 15 mg/ml DTT). Samples were analyzed by SDS–PAGE followed by immunoblotting with rabbit peroxidase-anti-peroxidase (PAP) antibodies (Z0113; Dako) or rabbit anti-GFP (ab6556; Abcam) and rabbit anti-Zwf1 (Miller et al, 2015) followed by goat anti-rabbit IgG-HRP (#111-035-003; Dianova) and imaging on an LAS-4000 system (Fuji).

## In vitro ubiquitination assays

6xHis-Rad6, 6xHis-Ubc4, Ubi-ProtA-6xHis, and Ubi$^{K48R}$-ProtA-6xHis were expressed in E.coli BL21(DE3) pRIL and purified over a prepacked HisTrap FastFlow column (GE Healthcare). FLAG-Ufd4, FLAG-Ubr1, FLAG-Ufd2, and FLAG-Tom1 were overexpressed in yeast from

a GPD promoter and purified as described (Hwang & Varshavsky, 2008; Hwang et al, 2009). Purified yeast Uba1 and ubiquitin were purchased from BostonBiochem (#E-300 and #U-100SC, respectively). Final protein concentrations were 100 nM (Uba1), 80 $\mu$M (ubiquitin), 1 $\mu$M (Rad6), 1 $\mu$M (Ubc4), 200 nM (Ubr1), 200 nM (Ufd4), 200 nM (Ufd2), 200 nM (Tom1), 125 nM (Ubi-ProtA), 125 nM (Ubi$^{K48R}$-ProtA), in 20 $\mu$l reactions containing 4 mM ATP (#1191, Merck), 150 mM NaCl, 5 mM MgCl2, 1 mM DTT, and 50 mM Hepes (pH 7.5). All reactions contained Uba1, ubiquitin, Rad6, and Ubc4. Reactions were pipetted on ice, incubated at 30°C for 30 min, quenched by addition of 20 $\mu$l 5× SDS sample buffer (50% [vol/vol] glycerol, 10% [wt/vol] SDS, 250 mM Tris–HCl, pH 6.8, 62.5 mM EDTA, and 5% [vol/vol] $\beta$-mercaptoethanol) and incubation at 95°C for 5 min, and analyzed using 4–12% NuPAGE Bis-Tris gradient gels (Invitrogen) followed by immunoblotting with rabbit peroxidase–anti-peroxidase (PAP) antibodies (Z0113; Dako) and imaging on a LAS-4000 system (Fuji).

### Fluorescence microscopy

Yeast strains were grown in SC medium at 23°C to ~8 × 10$^6$ cells/ml. Control strains not expressing fluorescent proteins and Tom1-GFP strains additionally expressing mCherry from a constitutive promoter were mixed 1:1 and attached to glass-bottom 96-well plates (MGB096-1-2-LG-L; Matrical) as described (Khmelinskii & Knop, 2014). Image stacks were acquired on a Nikon Ti-E wide field epifluorescence microscope with a 60× ApoTIRF oil immersion objective (1.49 NA; Nikon), an LED light source (SpectraX; Lumencor), a Flash4 sCMOS camera (Hamamatsu). Segmentation was performed in the bright-field channel using CellX (Mayer et al, 2013). Flat-field correction was performed using a reference image derived from a well containing recombinant mCherry-sfGFP fusion protein and average fluorescence across the stack was calculated for each cell. Cells were classified as autofluorescence control or sample separately for each field of view by fitting a two-component Gaussian mixture model to the mCherry intensity values and assigning each cell to the class with the higher posterior probability. GFP intensity of all control cells within a field of view was averaged and subtracted from the sample GFP intensities.

### Quantification of reporter transcripts by qPCR

Cells were grown at 23°C to a density of 1 × 10$^7$ cells/ml. RNA was isolated by hot phenol extraction as described (Collart & Oliviero, 1993). Reverse transcription was performed using Superscript IV First-Strand Synthesis System with a gene specific primer (tFT-RT: 5′-ATGCCTTTTCATATGGTCTGG-3′; TAF10-RT: 5′-CGCTACGGAAGACCTGATC-3′) according to manufacturer's instructions. Quantitative real-time PCR was performed using Roche LightCycler 480 SYBR Green I Master Mix (tFT-Primer: 5′-GCCAACCCTAGTAACAACTTTG-3′ & 5′-ATGCCTTTTCA-TATGGTCTGG-3′; TAF10-Primer: 5′-TAGCAGATGTACGAGTGAAACG-3′ & 5′-CGCTACGGAAGACCTGATC-3′) on a LightCycler 480 system according to the manufacturer's instructions. Cp-values were normalized to a reference gene (*TAF10*).

## Supplementary Information

## Acknowledgements

We acknowledge the support of Bernd He$\beta$ling (ZMBH Mass Spectrometry facility) and Monika Langlotz (ZMBH Flow Cytometry and FACS facility). We thank Birgit Besenbeck for technical support, Daniel Kirrmaier for support with fluorescence microscopy, and Frauke Melchior and Marius Lemberg for critically reading the manuscript. This work was supported by the Deutsche Forschungsgemeinschaft (SFB1036) to M Knop and an MSc/PhD fellowship from the HBIGS graduate school to I Kats.

## Author Contributions

I Kats: conceptualization, investigation, methodology, and writing—original draft.
C Reinbold: investigation, visualization, methodology, and writing—review and editing.
M Kschonsak: investigation.
A Khmelinskii: conceptualization, investigation, and methodology.
L Armbruster: investigation.
T Ruppert: investigation and methodology.
M Knop: conceptualization, supervision, funding acquisition, project administration, and writing—original draft, review, and editing.

## Conflict of Interest Statement

The authors declare that they have no conflict of interest.

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
