## [Reviewer comments · Life Science Alliance]

Life Science Alliance

Upregulation of Ubiquitin-Proteasome activity upon loss of NatA dependent N-terminal acetylation

Iliia Kats, Christian Reinbold, Marc Kschonsak, Anton Khmelinskii, Laura Armbruster, Thomas Ruppert, and Michael Knop
DOI: <https://doi.org/10.26508/lsa.202000730>

Corresponding author(s): Michael Knop, Heidelberg University

Review Timeline:

Submission Date:	2020-04-06
Editorial Decision:	2020-05-07
Revision Received:	2021-09-15
Editorial Decision:	2021-10-04
Revision Received:	2021-10-26
Accepted:	2021-10-27

Transaction Report:

May 7, 2020

Re: Life Science Alliance manuscript #LSA-2020-00730-T

Prof. Michael Knop
University of Heidelberg and German Cancer Research Center
ZMBH
Im Neuenheimer Feld 282
Heidelberg 69120
Germany

Dear Michael,

Thank you for submitting your manuscript entitled "Upregulation of Ubiquitin-Proteasome activity upon loss of NatA dependent N-terminal acetylation" to Life Science Alliance. The manuscript was assessed by expert reviewers, whose comments are appended to this letter. I am writing in lieu of LSA executive editor Andrea Leibfried, who recused herself on account of your future cooperation.

Briefly, N-terminal acetylation of proteins by either NatA, NatB or NatC was proposed previously as a degradation signal, but Nt-acetylation can also stabilize proteins. Nt-acetylated N-termini can be recognized and ubiquitinated by specific E3 ligases of the Ac/N-end rule pathway. You recently published a multiplexed protein stability (MPS) profiling system in yeast that allowed them to explore the degron propensity of different N-termini (Kats 2018, 29727619) and reported that reporters with a proline residue after the ubiquitin were degraded faster in cells deficient for NatA (Ubi-PZ-tFT). You here explore the mechanism further using tFT reporters from the Mol Cell paper. Your report that the enhanced degradation of these in the absence of NatA activity is due to several factors that act additively:

- Nt-acetylation by NatA promotes the ubiquitin-independent degradation of Rpn4.
- Tom1 is an E3 ligase in the UFD pathway and contributes to the degradation of the reporter. It catalyses K48 and K11 linkages. Tom1 expression is upregulated in NatA-deficient cells
- The overexpression of Ufd4 or Tom1 is sufficient to destabilize the PP reporter

Since the absence of NatA can still accelerate degradation of the UbiG76V-tFT reporter to some extent in cells that lack Ubr1, Ufd4, Tom1 and rpn4, you propose that the enhanced degradation is ultimately a combination of all these factors in concert with a generally higher UPS activity due to elevated Rpn4 levels.

All referees report potential interest of the research.

Ref #1 notes that the manuscript reports on 'several loosely connected mechanisms'. We concur with the referee requests for overexpression of NAA10. We recognize the in vitro experiment suggested is likely non-trivial, although the basic protocol appears to be published: we will not require addition of this experiment, although it would strengthen the paper if the data were available; The issue on fig 2G has to be addressed by redoing the experiment if necessary. Please address the referee concern in the text that main assay may underestimate co-translational degradation.

Ref #2: notes the generally small effects and argues that the biological consequences remain unclear. The referee raises a list of issues that should be addressed in the manuscript text: pt. 2, 4, 5, 6, 8, 14, 15. In addition, we invite a response to pt. 3, 10, 12, 13 and suggest that pt 7 requires toning down the conclusions. Specific issues that should be addressed experimentally: pt.: 1 fig 1 minimally needs another control for synthesis. ; pt. 9, 11: controls have to be added.

We judge the required revision to be realistic and valuable and this invite a resubmission of a revised manuscript

To upload the revised version of your manuscript, please log in to your account: <https://lsa.msubmit.net/cgi-bin/main.plex>
You will be guided to complete the submission of your revised manuscript and to fill in all necessary information. Please get in touch in case you do not know or remember your login name.

Thank you for this interesting contribution to Life Science Alliance. We are looking forward to receiving your revised manuscript.

best wishes,

Bernd

Bernd Pulverer
Head of Scientific Publications
EMBO

B. MANUSCRIPT ORGANIZATION AND FORMATTING:

Reviewer #1 (Comments to the Authors (Required)):

In this report, Kats et al. demonstrated that degradation of model UFD substrates was increased in NatA (naa10) mutant. The authors provided several mechanisms to explain this observation. While these putative mechanisms remain loosely connected, this study adds to our understanding of the complexity of the UFD pathway and the pleiotropic effects of Nt-acetylation mediated by NatA. This report should be interesting to scientists in the field of protein homeostasis. The reviewer has several comments for the authors to consider for their revised manuscript.

1. This study only measured the effect of deletion of NAA10 on the degradation of model UFD substrates and Rpn4(1-80)-tFT reporters. It is important to also examine the effect of overexpression of NAA10. For example, in Figure 1 the effect of deletion of NAA10 on the stability of UbiG76V-eK-tFT is marginal in UFD4 wt background, suggesting that the E3 play a dominant role. Yet, deletion of NAA10 substantially destabilizes UbiG76V-eK-tFT in the absence of UFD4, implying that the effect of NAA10 may be independent of UFD4. It would be interesting to see if overexpression of NAA10 slows the degradation of UbiG76V-eK-tFT in UFD4 wt background.

2. The authors hypothesized that Nt-acetylation of Rpn4 affects its N-terminal ubiquitin-independent degron, i.e., Nt-acetylation enhances RPN4 degradation via Ub-independent degron. Is it possible to directly test this hypothesis using an in vitro reconstitution degradation system? It has been shown that Rpn4(1-80) is degraded by purified 26S proteasome in vitro. Can the authors prepare acetylated Rpn4(1-80) and compare its degradation with non-acetylated counterpart in vitro?

3. In Figure 2G, the authors assessed if Rpn4 mediates the accelerated degradation of the UbiG76V-tFT reporter in the naa10Δ mutant carrying the rpn4A2N allele. This experiment design is a bit strange because rpn4A2N is acetylated by NatB but not NatA. Its steady-state level, presumably regulated by Nt-acetylation, is not expected to be noticeably affected by deletion of NAA10. Should this assay be done in a NatB mutant?

4. This study is heavily relied on the tFT technique developed by the authors in their previous studies. This assay is based on differential folding efficiencies of mCherry and sfGFP, and measures the ratio of mCherry/sfGFP as a function of degradation. This technique may work beautifully for posttranslational degradation, but likely underestimates cotranslational degradation

because proteasomal degradation of nascent polypeptides is processive. A large portion of UFD substrates is degraded cotranslationally.

Reviewer #2 (Comments to the Authors (Required)):

Summary of paper

Kats et al. studied the effect of N-terminal acetylation on protein stability in yeast. Their previous study (Kats et al. Mol. Cell 2018) revealed accelerated turnover of UFD substrates in cells lacking the N-terminal acetyltransferase NatA. This observation was investigated further here and revealed additional details of how N-terminal acetylation modulates the ubiquitin-proteasome system. They found that N-terminal acetylation of Rpn4 contributes to its ubiquitin-independent degradation. Rpn4 is a transcription factor that up-regulates proteasome gene transcription. Loss of NatA leads to more Rpn4 in cells but no obvious change in proteasome expression levels. In addition, Kats et al. demonstrate that the E3 ligase Tom1 contributes to UFD substrate ubiquitination, in addition to the previously identified Ufd2, Ufd4 and Ubr1 E3s, using a combination of in vitro and in vivo approaches. They propose a plausible explanation (upregulation of E3 ligases) for how deletion of NatA accelerates the turnover of UFD substrates. Overall, the study was carefully designed and most of the results are convincing. The effects are small, and in some cases, such as the Rpn4 degradation effect, the biological consequences are not obvious. While the observations here will be of some interest to researchers in the UPS field, they might be of limited appeal to a wider audience.

Specific comments/questions/suggestions that should be addressed:

- 1) In the CHX-chase experiment shown in Supplemental Fig 1, the degradation rates actually appear faster in the *ubr1 ufd4* double mutant compared to the *ubr1 ufd4 naa10* triple mutant. Although protein abundance is clearly reduced in the triple mutant, one could argue *naa10Δ* is influencing synthesis levels.
- 2) Fig 2E - "Label-free quantitative mass spectrometry of full-length Rpn4 confirmed NatA-dependent acetylation of Rpn4 and NatB-dependent acetylation of Rpn4A2N (Fig 2E)." The data presented in this figure should be clarified or labeled differently. Was mass spec performed on both Rpn4 and Rpn4A2N? As shown, I would assume the experiment was performed only on Rpn4. If mass spec was not performed on Rpn4A2N, I don't believe the assumption could be made that "NatB-dependent acetylation of Rpn4A2N" was confirmed (perhaps "supported").
- 3) Fig 2F - Data with Rpn4Δ(211-229)-TAP is consistent with experiments done on Rpn4(1-80)-tFT. Could you comment on why deletion of NatB doesn't appear to affect Rpn4A2N,Δ(211-229)-TAP and Rpn4(1-80)A2N in a similar manner?
- 4) There is no figure legend for Supplemental Figure 2C. This was a pretty sloppy document in general. Figure panels mixed up (see below) and also many incomplete citations in the text.
- 5) "Moreover, only mono- and diubiquitinated species were seen in the *tom1Δ* mutant, when using the UbiK48R,G76V-tFT reporter, despite strong polyubiquitination of this reporter in wild type cells."
I would suggest revising this to "mainly."
- 6) "Moreover, detailed analysis of the banding pattern revealed that in the presence of Tom1 tri-ubiquitinated species of different apparent molecular weight were generated (Fig 4E), indicating that ubiquitin conjugates synthesized by Tom1 and Ufd4 are clearly distinct."
I believe this sentence should be re-worded for ease of understanding.
- 7) "Levels of both E3 ligases were elevated in *naa10Δ* cells compared to wild type (Fig 5A and Fig S3)."
Based on the western blots shown in Supplemental Figure 3, it's difficult to see any change in protein levels. I believe this statement lacks sufficient evidence. If authors decide to retain these data, I feel that the western blot should be shown in the main text. Also, I believe the appropriate figure being referenced here is Fig 5B.
- 8) "The Ubi-PP-tFT reporter was more stable in the wild type background but was only weakly destabilized in a strain overexpressing Ubr1 (Fig 5B), consistent with a negligible contribution of Ubr1 to UFD in vivo (Figs 1C and 3C (Hwang et al., 2010a)). However, overexpression of Ufd4 or Tom1 strongly destabilized the PP reporter (Fig 5B). Moreover, these results suggest that increased abundance of UFD E3 ligases could explain accelerated turnover of UFD substrates in the *naa10Δ* mutant."
The appropriate figure being referenced here is Fig 5C.
- 9) The data presented strongly supports the involvement of Tom1 in mediating accelerated turnover of UFD substrates; however, I strongly recommend at least one cycloheximide chase is shown to verify stabilization of Ub-FT in *tom1Δ* cells.
- 10) The in vitro experiments presented suggest Tom1 targets UFD substrates after UFD4 function. However, the in vivo

experiments shown demonstrating UFD stabilization by Tom1 were done in *ufd4 ubr1* double mutants. Could you comment or speculate on how these proteins are being targeted for degradation in the absence of Ufd4?

- 11) Given Tom1 weakly binds the negative control EH-tFT (Fig 3B), it might be worth verifying this protein is not degraded by Tom1.
- 12) The abstract mentions Tom1 is involved in UFD turnover by forming K11 and K29 ubiquitin branches, yet the data presented in 4F implies Tom1 can also form K48 branches in vitro.
- 13) Why doesn't *naa20D* affect *rpn4A2N* degradation (Fig. 2F)? This would be predicted.
- 14) Last paragraph of Results should be part of Discussion.
- 15) The ubiquitination analysis with Tom1, etc. is densely written. A careful rewrite is strongly suggested. Phrases such as "a markedly reduced acceleration" in "In the *ubr1Δ ufd4Δ* background, cells lacking Tom1 showed a markedly reduced acceleration of UbiG76V-tFT reporter degradation upon deletion of NAA10 compared to Tom1-proficient cells..." can be simplified.

Reviewer #3 (Comments to the Authors (Required)):

The manuscript is based on the hypothesis that n terminal acetylation impact on the presentation of n-terminal degron and the degradation of corresponding proteins. However, there are contrasting evidence already on this in the literature. I found the manuscript quite difficult to follow and the main finding of the manuscript is lost in numerous speculation and assumptions. The manuscript needs a significant reorganisation before it can be considered for publication. It could be important to provide some model which identifies clearly the main finding of the manuscript and the underlying hypothesis.

1. This study only measured the effect of deletion of NAA10 on the degradation of model UFD substrates and Rpn4(1-80)-tFT reporters. It is important to also examine the effect of overexpression of NAA10. For example, in Figure 1 the effect of deletion of NAA10 on the stability of UbiG76V-eK-tFT is marginal in UFD4 wt background, suggesting that the E3 play a dominant role. Yet, deletion of NAA10 substantially destabilizes UbiG76V-eK-tFT in the absence of UFD4, implying that the effect of NAA10 may be independent of UFD4. It would be interesting to see if overexpression of NAA10 slows the degradation of UbiG76V-eK-tFT in UFD4 wt background.

→ Overexpression of NAA10 did not have an effect on reporter stability in the absence of UFD4. Endogenous levels of NAA10 seem to have already maximal acetylating effect on Rpn4. This data is shown in Fig. S1D and commented the last paragraph of Results Chapter 'NatA affects turnover of UFD substrates'

2. The authors hypothesized that Nt-acetylation of Rpn4 affects its N-terminal ubiquitin-independent degron, i.e., Nt-acetylation enhances RPN4 degradation via Ub-independent degron. Is it possible to directly test this hypothesis using an in vitro reconstitution degradation system? It has been shown that Rpn4(1-80) is degraded by purified 26S proteasome in vitro. Can the authors prepare acetylated Rpn4(1-80) and compare its degradation with non-acetylated counterpart in vitro?

This would indeed be an interesting hypothesis to follow up. But this would involve a significant amount of work that goes beyond the scope of his manuscript.

3. In Figure 2G, the authors assessed if Rpn4 mediates the accelerated degradation of the UbiG76V-tFT reporter in the naa10Δ mutant carrying the rpn4A2N allele. This experiment design is a bit strange because rpn4A2N is acetylated by NatB but not NatA. Its steady-state level, presumably regulated by Nt-acetylation, is not expected to be noticeably affected by deletion of NAA10. Should this assay be done in a NatB mutant?

We agree that one could also investigate this question in the natB mutant and compare the effect of the N-terminal modification of Rpn4 on degradation of the reporter. We now included this experiment into Figure 2G and we found a weak acceleration of degradation of the reporter in the AN-Rpn4/nat20 background compared to the wt-Rpn4/nat20 background. This is consistent with the idea that N-terminal Rpn4 acetylation contribute to the degradation of UFD substrates.

To introduce this data, we rewrote parts of this paragraph (green text):

To assess if N-acetylation of Rpn4 mediates the accelerated degradation of the UbiG76V-tFT reporter, we measured turnover of this reporter in cells naa20Δ mutant cells expressing wild type Rpn4 or the AN-Rpn4 mutant. We found a weak acceleration of degradation of the reporter in the AN-Rpn4/naa20Δ background compared to the wild type-Rpn4/naa20Δ background (Fig 2G). This is consistent with the idea that N-terminal Rpn4 acetylation contributes to the degradation of UFD substrates.

In addition to this we optimized the description of the data in 2D and E, where the AN-Rpn4 mutant is introduced, to help a reader to better understand this rather complex part of the data.

4. This study is heavily relied on the tFT technique developed by the authors in their previous studies. This assay is based on differential folding efficiencies of mCherry and sfGFP, and measures the ratio of mCherry/sfGFP as a function of degradation. This technique may work beautifully for posttranslational degradation, but likely underestimates cotranslational degradation because proteasomal degradation of nascent polypeptides is processive. A large portion of UFD substrates is degraded cotranslationally.

We agree with the reviewer that the tFT may underestimate cotranslational degradation. However, the sfGFP protein used here is resistant to processive degradation, resulting in a fluorescent proteasomal escape product (Khmelniskii et al. 2015). We observe substantial amounts of this escape product even in wild type yeast (Fig. 4A, B). Since we are only concerned

with relative changes in degradation rate between conditions and not absolute half-lives, we believe that the tFT method still offers sufficient sensitivity for our purposes.

Rev. 2

1) In the CHX-chase experiment shown in Supplemental Fig 1, the degradation rates actually appear faster in the *ubr1 ufd4* double mutant compared to the *ubr1 ufd4 naa10* triple mutant. Although protein abundance is clearly reduced in the triple mutant, one could argue *naa10Δ* is influencing synthesis levels.

Indeed, the differences in protein levels are striking and may not be explained completely by differences in protein stability. To investigate this we used qPCR to quantify the abundance of the mRNA of the reporter (→ new panel S1B). We indeed found that deletion of *NatA* led to an approx. 40% reduction of the mRNA level of the reporter. This indicates that idea of transcription levels being influenced by *naa10Δ* seems to be correct. This might explain the lowered protein abundance that is visible in the Western Blot analysis. Global changes of changed of transcription levels in *NatA* mutant were also reported in another study (Friedrich et al., 2019).

We repeated the CHX-chase multiple times to gain more robust data for better quantification. Quantification of the experiments detected a slightly decreased half-life of the reporter in the *naa10* mutant (→ new panel S1C). This data is shown in the supplement and we changed the text accordingly.

2) Fig 2E - "Label-free quantitative mass spectrometry of full-length Rpn4 confirmed *NatA*-dependent acetylation of Rpn4 and *NatB*-dependent acetylation of Rpn4A2N (Fig 2E)." The data presented in this figure should be clarified or labeled differently. Was mass spec performed on both Rpn4 and Rpn4A2N? As shown, I would assume the experiment was performed only on Rpn4. If mass spec was not performed on Rpn4A2N, I don't believe the assumption could be made that "*NatB*-dependent acetylation of Rpn4A2N" was confirmed (perhaps "supported").

We are fully aware of this. As a matter of fact, we have performed the ms experiment also with Rpn4A2N, but we forgot to reference this data in the text, which was actually shown in Fig. S2C. We now reference this panel in the text.

3) Fig 2F - Data with Rpn4Δ(211-229)-TAP is consistent with experiments done on Rpn4(1-80)-tFT.

Could you comment on why deletion of *NatB* deletion doesn't appear to affect Rpn4A2N,Δ(211-229)-TAP and Rpn4(1-80)A2N in a similar manner?

Half-lives in Fig. 2F were estimated based on CHX chases, whereas stabilization of Rpn4(1-80)-tFT in Fig. 2D in *naa20Δ* was determined based on flow cytometry using the tFT. CHX chases are inherently more difficult to quantify than flow cytometry. Moreover, the *naa20Δ* strains are quite sick and difficult to work with. As can be seen in Fig. 2F, the half-life estimates vary greatly between replicate Western blots, making it difficult to assess the presence or absence of an effect in this experiment. We now discuss this briefly in the results section. We introduced the following explanation:

"Similarly, we expected that inactivation of *NatB* should also lead to a stabilization of Rpn4A2NΔ(211-229)-TAP, but the results were less clear because of large variation between individual replicates, probably due to the rather severe growth defect caused by the absence of *NatB*."

4) There is no figure legend for Supplemental Figure 2C. This was a pretty sloppy document in general. Figure panels mixed up (see below) and also many incomplete citations in the text.

Indeed, this legend is missing and we also forgot to reference this panel, (see point 2)

Legend and citation were adjusted.

[Figure removed by editorial staff per authors' request]

(C) Quantification of the ion chromatograms of Nt-acetylated and unmodified N-terminal peptides derived from full-length Rpn4 variants obtained by label-free mass spectrometry. The data shows the ratio of acetylated and unmodified peptides for the different strains and Rpn4 variants, as indicated. + indicates wild type, Δ indicates a deletion of the gene.

5) "Moreover, only mono- and diubiquitinated species were seen in the tom1 Δ mutant, when using the UbiK48R,G76V-tFT reporter, despite strong polyubiquitination of this reporter in wild type cells."

I would suggest revising this to "mainly."

Wording was changed.

6)" Moreover, detailed analysis of the banding pattern revealed that in the presence of Tom1 tri-ubiquitinated species of different apparent molecular weight were generated (Fig 4E), indicating that ubiquitin conjugates synthesized by Tom1 and Ufd4 are clearly distinct."

I believe this sentence should be re-worded for ease of understanding.

Indeed. Sentence was rephrased.

Moreover, detailed analysis of the banding pattern revealed that ubiquitin conjugates synthesized by Tom1 and Ufd4 are clearly distinct (Fig 4E). Tri-ubiquitinated species of a slightly smaller molecular weight were only generated in the presence of Tom1 and not Ufd4 alone.

7) "Levels of both E3 ligases were elevated in naa10 Δ cells compared to wild type (Fig 5A and Fig S3). "

Based on the western blots shown in Supplemental Figure 3, it's difficult to see any change in protein levels. I believe this statement lacks sufficient evidence. If authors decide to retain these data, I feel that the western blot should be shown in the main text. Also, I believe the appropriate figure being referenced here is Fig 5B.

Sentence was rephrased.

8) "The Ubi-PP-tFT reporter was more stable in the wild type background but was only weakly destabilized in a strain overexpressing Ubr1 (Fig 5B), consistent with a negligible contribution of Ubr1 to UFD in vivo (Figs 1C and 3C (Hwang et al., 2010a)). However, overexpression of Ufd4 or Tom1 strongly destabilized the PP reporter (Fig 5B). Moreover, these results suggest that increased abundance of UFD E3 ligases could explain accelerated turnover of UFD substrates in the naa10 Δ mutant."

The appropriate figure being referenced here is Fig 5C.

Reference was adjusted.

9) The data presented strongly supports the involvement of Tom1 in mediating accelerated turnover of UFD substrates; however, I strongly recommend at least one cycloheximide chase is shown to verify stabilization of Ub-FT in tom1 Δ cells.

We now include in the Supplement (Fig S5A) an immune blot where we show that deletion of Tom1 leads to a clear stabilization of Ubi-G67V (+/-K48R)-tTF reporter constructs.

10) The in vitro experiments presented suggest Tom1 targets UFD substrates after UFD4 function. However, the in vivo experiments shown demonstrating UFD stabilization by Tom1 were done in *ufd4 ubr1* double mutants. Could you comment or speculate on how these proteins are being targeted for degradation in the absence of Ufd4?

UFD reporters already possess a non-cleavable Ubi-moiety that might be weakly recognized by various other ubiquitin ligases that can lead to residual degradation in the absence of Ufd4.

11) Given Tom1 weakly binds the negative control EH-tTF (Fig 3B), it might be worth verifying this protein is not degraded by Tom1.

We did a tTF-assay using Ubi-EH-tTF and found not dependency of Tom1. This data was included in Fig S5B.

12) The abstract mentions Tom1 is involved in UFD turnover by forming K11 and K29 ubiquitin branches, yet the data presented in 4F implies Tom1 can also form K48 branches in vitro.

Text was adjusted.

13) Why doesn't *naa20D* affect *rpn4A2N* degradation (Fig. 2F)? This would be predicted.

See our response to point 3.

14) Last paragraph of Results should be part of Discussion.

Paragraph was moved to the discussion section.

15) The ubiquitination analysis with Tom1, etc. is densely written. A careful rewrite is strongly suggested. Phrases such as "a markedly reduced acceleration" in "In the *ubr1Δ ufd4Δ* background, cells lacking Tom1 showed a markedly reduced acceleration of UbiG76V-tTF reporter degradation upon deletion of NAA10 compared to Tom1-proficient cells..." can be simplified.

Text was adjusted.

October 4, 2021

RE: Life Science Alliance Manuscript #LSA-2020-00730-TR

Prof. Michael Knop
Heidelberg University
ZMBH
Im Neuenheimer Feld 282
Heidelberg 69120
Germany

Dear Dr. Knop,

Thank you for submitting your revised manuscript entitled "Upregulation of Ubiquitin-Proteasome activity upon loss of NatA dependent N-terminal acetylation". We would be happy to publish your paper in Life Science Alliance pending final revisions necessary to meet our formatting guidelines.

- please upload your main and supplementary figures as single files
- please add the Twitter handle of your host institute/organization as well as your own or/and one of the authors in our system
- please make sure the author order in your manuscript and our system match
- please add your main, supplementary figure, and table legends to the main manuscript text after the references section
- please consult our manuscript preparation guidelines <https://www.life-science-alliance.org/manuscript-prep> and make sure your manuscript sections are in the correct order
- please upload your Tables in editable .doc or excel format
- please use the [10 author names, et al.] format in your references (i.e. limit the author names to the first 10)
- please add callouts for Figure S3A-C to your main manuscript text
- please make sure there are sizes indicated next to each blot

A. FINAL FILES:

B. MANUSCRIPT ORGANIZATION AND FORMATTING:

Sincerely,

Reviewer #1 (Comments to the Authors (Required)):

After reviewing the changes/modifications incorporated in the resubmission, the reviewer think that the revised manuscript now warrants publication in LSA.

Reviewer #2 (Comments to the Authors (Required)):

In general, the responses to my specific comments were good, and some of the newer experiments clarified my earlier concerns.

I guess the bottom line, which was emphasized in the Friedrich et al. paper published earlier this year in Cell Reports (cited here; Dr. Knop is a co-author), is that N-terminal acetylation has a very small direct impact on general protein degradation by the ubiquitin system in *S. cerevisiae*. This is confirmed in the current paper with several examples, such as the changes in Rpn4 levels, where changes in its synthesis appear to more strongly affect its levels than does the small increase in its half-life when NatA is deleted. I believe these are real effects on Rpn4 stability and levels of certain E3s due to NatA, for sure, but they are frustratingly small.

This is a judgement call for the editor. As a referee, I feel the data are solid and the experiments are well done. The question is really about the significance of the advance, particularly given the very small degree of ubiquitin-proteasome system upregulation observed and the partial overlap with the conclusions from the Friedrich et al. paper.

October 27, 2021

RE: Life Science Alliance Manuscript #LSA-2020-00730-TRR

Prof. Michael Knop
Heidelberg University
ZMBH
Im Neuenheimer Feld 282
Heidelberg 69120
Germany

Dear Dr. Knop,

Thank you for submitting your Deleted entitled "Upregulation of Ubiquitin-Proteasome activity upon loss of NatA dependent N-terminal acetylation". It is a pleasure to let you know that your manuscript is now accepted for publication in Life Science Alliance. Congratulations on this interesting work.

DISTRIBUTION OF MATERIALS:

Again, congratulations on a very nice paper. I hope you found the review process to be constructive and are pleased with how the manuscript was handled editorially. We look forward to future exciting submissions from your lab.

Sincerely,
